# Non-equilibrium dynamics of spin-lattice coupling

Hiroki Ueda [1,2] ✉, Roman Mankowsky [1], Eugenio Paris [1], Mathias Sander [1], Yunpei Deng[1], Biaolong Liu[1], Ludmila Leroy[2], Abhishek Nag [1], Elizabeth Skoropata [2], Chennan Wang [3], Victor Ukleev[2,6], Gérard Sylvester Perren [2], Janine Dössegger [4], Sabina Gurung[4], Cristian Svetina [1,7], Elsa Abreu [4], Matteo Savoini[4], Tsuyoshi Kimura[5], Luc Patthey[1], Elia Razzoli[1], Henrik Till Lemke [1], Steven Lee Johnson [1,4] & Urs Staub [2] ✉

Quantifying the dynamics of normal modes and how they interact with other excitations is of central importance in condensed matter. Spin-lattice coupling is relevant to several sub-fields of condensed matter physics; examples include spintronics, high-$T_c$ superconductivity, and topological materials. However, experimental approaches that can directly measure it are rare and incomplete. Here we use time-resolved X-ray diffraction to directly access the ultrafast motion of atoms and spins following the coherent excitation of an electromagnon in a multiferroic hexaferrite. One striking outcome is the different phase shifts relative to the driving field of the two different components. This phase shift provides insight into the excitation process of such a coupled mode. This direct observation of combined lattice and magnetization dynamics paves the way to access the mode-selective spin-lattice coupling strength, which remains a missing fundamental parameter for ultrafast control of magnetism and is relevant to a wide variety of materials.

Crystals with magnetic order have dynamics governed largely by normal modes of lattice and spin displacement, giving rise to quasiparticles such as phonons and magnons. The frequencies of the modes are defined by the microscopic interaction among individual atoms or among the magnetic moments localized at magnetic ions. These excitations are commonly studied by spectroscopic techniques such as inelastic neutron scatterings, X-ray scatterings, and Raman spectroscopy. In some cases, direct measurements of coherent excitations in the time domain have become possible, for example by performing pump-probe measurements of coherent vibrational and spin dynamics[1,2].

The coupling between spins and lattice is a fundamental interaction important to a wide variety of magnetic and electronic phenomena. Examples are ultrafast energy and angular-momentum transfer[3] and electric control of magnetism[4], relating to efficient energy transfer and storage application. This coupling can also lead to hybrid normal modes that involve both lattice and spin dynamics.

In this study, we directly observe how the spins and lattice interact as components of a hybrid normal mode. We focus on a specific multiferroic where the spin-lattice coupling is the origin of the magnetoelectric effect and responsible for non-reciprocal transport[5]. The breaking of inversion symmetry by magnetic order can cause magnetic

[1]SwissFEL, Paul Scherrer Institute, 5232 Villigen-PSI, Switzerland. [2]Swiss Light Source, Paul Scherrer Institute, 5232 Villigen-PSI, Switzerland. [3]Départment de Physique and Fribourg Center for Nanomaterials, Université de Fribourg, 1700 Fribourg, Switzerland. [4]Institute for Quantum Electronics, Physics Department, ETH Zurich, 8093 Zurich, Switzerland. [5]Department of Advanced Materials Science, University of Tokyo, Kashiwa, Chiba 277-8561, Japan. [6]Present address: Helmholtz-Zentrum Berlin für Materialien und Energie, Albert-Einstein-Straße 15, 12489 Berlin, Germany. [7]Present address: Madrid Institute for Advanced Studies, IMDEA Nanociencia, Ciudad Universitaria de Cantoblanco, Calle Faraday 9, Madrid 28049, Spain. ✉e-mail: hiroki.ueda@psi.ch; urs.staub@psi.ch

excitations to acquire an additional polar phonic character, making them susceptible to the electric field component of light. The characteristic hybrid mode that arises from the first-order coupling between magnetism and ferroelectricity is called electromagnon[6] and provides a platform to us to investigate spin-lattice coupling in this material.

Electromagnons were originally envisioned as excitations arising from the coupling between magnetic order and the crystal lattice that gives rise to the equilibrium electric polarization, i.e., THz-induced oscillations of static electric polarization produced by magnetic order[7]. We call this type of electromagnons type I, following the nomenclature introduced by ref. 8. While type I electromagnons have been reported[8], for many materials magnetostrictive coupling to a polar lattice distortion gives rise to electromagnons with much larger oscillator strengths[9–11]. In these "type II" electromagnons[8] the polar lattice distortion modifies exchange interactions, resulting in a magnetic response. Coherent excitations of these hybrid modes with pulsed electric fields hold great potential to manipulate magnetism on an ultrafast timescale, as theoretically predicted for TbMnO$_3$[12]. While the spin dynamics of coherently excited electromagnons have been observed by time-resolved magnetic diffraction[13–15], the underlying structural dynamics of electromagnons have not yet been directly observed in any material so far.

The Y-type hexaferrites (Ba,Sr)$_2$Me$_2$(Fe,Al)$_{12}$O$_{22}$ (Me: transition metal ion) belong to the family of magnetoelectric multiferroics with space group $R\bar{3}m$ and a large lattice parameter along [001] (≈43.3 Å), as shown in Fig. 1a. Their magnetic structure is commonly described as a magnetic block structure, where each magnetic block, termed S or L, contains a certain number of collinearly coupled Fe magnetic moments[16]. The Fe-O-Fe bond angles at the boundary between adjacent magnetic blocks create magnetic frustration. As a result of the magnetic frustration, the Y-type hexaferrite investigated here exhibits the transverse-conical magnetic structure, as displayed in Fig. 1b. The magnetic structure can be described as a combination of a ferrimagnetic component and a spin-spiral component. While the ferrimagnetic component hosts net magnetization, the spin-spiral component gives

rise to electric polarization via static spin-lattice interaction[16]. This makes the transverse-conical phase multiferroic.

A clear electric-dipole active resonance in the THz absorption spectra has been reported in Y-type hexaferrites for electric fields ($E_{THz}$) parallel to [001][17,18]. Based on a detailed investigation of the temperature, magnetic-field, and $E_{THz}$-direction dependences of the THz optical properties[17,18], this mode has been attributed to an electromagnon mode. Out-of-phase oscillation of the magnetic moments towards [001] in the magnetic blocks can result in a transient electric dipole moment along [001] via magnetostriction, as schematically illustrated in Fig. 1c[17]. Resonant excitation of electromagnons by an intense THz pulse is known as an effective pathway to create large amplitude dynamic modulations of magnetic moments[13]. The large resonator strength found in Y-type hexaferrites implies a strong spin-lattice coupling, which is promising for possible ultrafast control of magnetism. Although the theory of these electromagnons is well developed, experimental observation of the vibrational component of electromagnons is still missing. The measurement of this structural component in combination with the magnetic response is of fundamental importance for experimentally demonstrating how electromagnons are excited following a THz pulse in a multiferroic. Moreover, it has the potential to directly determine the magnetoelectric (spin-lattice) coupling strength of the excitation.

## Results

Here we utilize time-resolved hard X-ray diffraction (tr-XRD) and time-resolved resonant soft X-ray magnetic diffraction (tr-RSXD) with few-cycle phase-stable THz excitations to observe the lattice and sublattice magnetization dynamics of electromagnons, respectively, in the multiferroic Y-type hexaferrite Ba$_{1.3}$Sr$_{0.7}$CoZnFe$_{11}$AlO$_{22}$. Clear oscillations in the time dependence of the (0 0 24) reflection intensity show the lattice component of this mode, whereas the oscillations of the (0 0 4.5) magnetic reflection intensity give a direct measure of the induced spin motion. Observed redshifts concomitant with increased damping at elevated temperatures confirm the identification of the oscillations as being due to the electromagnon. A direct comparison of magnetic and

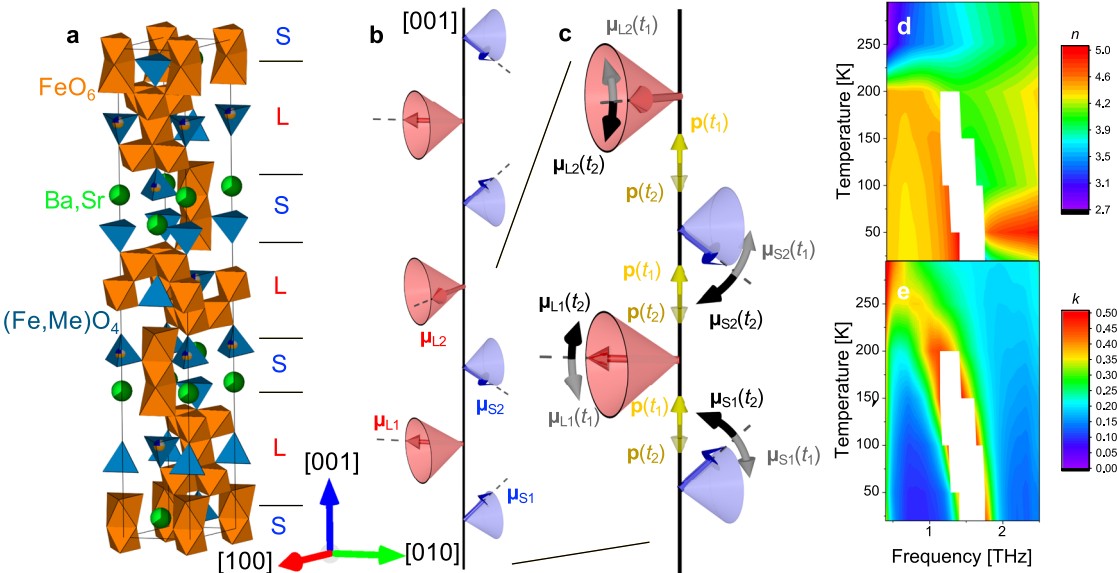

**Fig. 1 | Basic structures/characterization of Y-type hexaferrite. a** Crystal structure and **b** magnetic structure of the Y-type hexaferrite Ba$_{1.3}$Sr$_{0.7}$CoZnFe$_{11}$AlO$_{22}$. The magnetic structure is depicted by the representative magnetic moments of an S block and L block [μ$_S$ (blue arrow) and μ$_L$ (red arrow), respectively], which alternatively stack along [001]. **c** Proposed spin dynamics of the electromagnon in a Y-type hexaferrite:[17] snapshots of the dynamics at different times, $t_1$ (phase 0) and $t_2$ (phase π). A pair of the same adjacent family of magnetic moments (μ$_{S1}$ and μ$_{S2}$, and

μ$_{L1}$ and μ$_{L2}$) show anti-phase oscillations towards [001]. Such oscillations give rise to transient electric dipole moments along [001] through magnetostriction. **d** Temperature dependence of the real part, $n$, and **e**, the imaginary part, $k$, of the refractive index $n+ik$ in the THz range, measured by THz-TDS. The white region is a frequency range where no transmission through the sample is detected because of strong absorption. The crystal structure was drawn by VESTA[37].

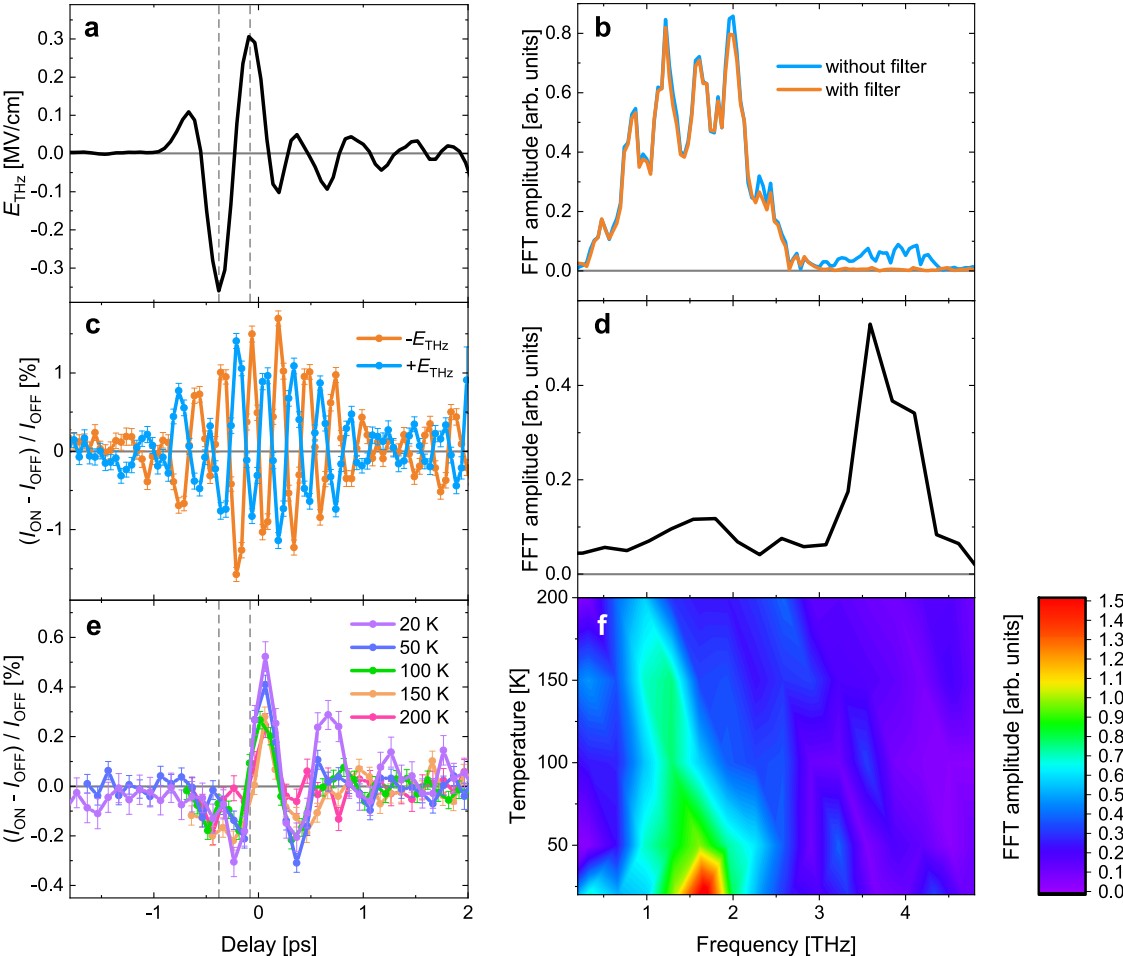

**Fig. 2 | tr-XRD signals. a** The incident THz pump pulse measured with electro-optic sampling at the sample position (see Methods for detail). **b**, THz pulse spectra, where blue and orange curves indicate data without and with the low-pass filter, respectively. **c** The (0 0 24) XRD intensity at 20 K as a function of pump-probe delay. The blue data show the XRD response for $+E_{THz}$, while the orange data show the response when inverting the phase of the THz electric field ($-E_{THz}$). **d** The FFT spectrum of the diffraction response. **e** Temperature dependence of the time-dependent diffraction intensity after insertion of a low-pass filter that cuts off components above 3 THz. **f** The FFT spectra of (**e**). Dashed lines in (**a**) and (**e**) highlight the phase shift of $\pi/2$ between the incident THz pulse and the (0 0 24) XRD intensity.

lattice responses opens the door for accessing mode-selective spin-lattice interactions important for ultrafast control of materials, and allows us to discuss how the two characters couple on ultrafast timescales.

## Lattice dynamics

THz time-domain spectroscopy (THz-TDS) data on the Y-type hexaferrite show a clear divergent feature centered at -1.7 THz ($T = 20$ K) in the imaginary part of the refractive index (see Fig. 1e). Upon heating, the mode shows a red shift concomitant with a weakening of the resonator strength. These results are consistent with the previous literature identifying the mode as an electromagnon[17,18]. Figure 2a shows a time trace of the electric field of the incident THz pulse ($E_{THz}$) in the tr-XRD experiment, whose direction was rotated to maximize the component along [001] (see Methods). The corresponding time trace of the (0 0 24) reflection intensity taken at a sample temperature of 20 K is shown in Fig. 2c. The figure also shows measurements of the time-dependent X-ray diffraction for opposite phases of the THz pulse ($+E_{THz}$ and $-E_{THz}$). Figure 2b, d shows the corresponding Fast Fourier Transformation (FFT) amplitudes of the incident THz pulse and the (0 0 24) reflection intensity, respectively. As described in Methods, the photon energy used for the experiment is 7.5 keV, slightly above the Fe $K$ edge, where the large imaginary part of the X-ray scattering factor of Fe enables us to detect polar motions. The (0 0 24) reflection was chosen because it is exclusively sensitive to atomic motions along

[001] and it allows us to use a large incidence angle of a collinear THz and X-ray pulses to the sample surface, resulting in a large $E_{THz}$ inside the sample. This reflection also makes possible a grazing exit angle of 0.5° that results in a short probing depth which is more homogeneously pumped (See Methods for the penetration depths of the beams). Furthermore, with respect to a given atomic motion, the relative change of diffraction intensities is generally large for a reflection with a small structure factor, as is the case for the (0 0 24) reflection. Different reflections will therefore also be sensitive to additional modes. (Supplementary Note I) Clear oscillations are observed in the diffraction intensities. These dynamics change sign when reversing the $E_{THz}$ direction, unequivocally indicating coherent dipole-active lattice vibrations. The FFT of the time traces reveals two peaks at frequencies 1.7 THz and 3.6 THz, with the latter having an amplitude about five times *larger* than the former. The lower frequency matches that of the electromagnon resonance, while the higher frequency corresponds to an infrared-active optical phonon mode[18]. Note that the spectral power of the incident THz pulse at 3.6 THz is approximately an order of magnitude *weaker* than that around 1.7 THz, as shown in Fig. 2b.

To exclude that the observation of the higher-frequency mode represents a phonon upconversion process caused by non-linear phonon interactions as observed in SrTiO$_3$[19], a low-pass filter has been inserted to fully suppress the weak high-frequency component in the

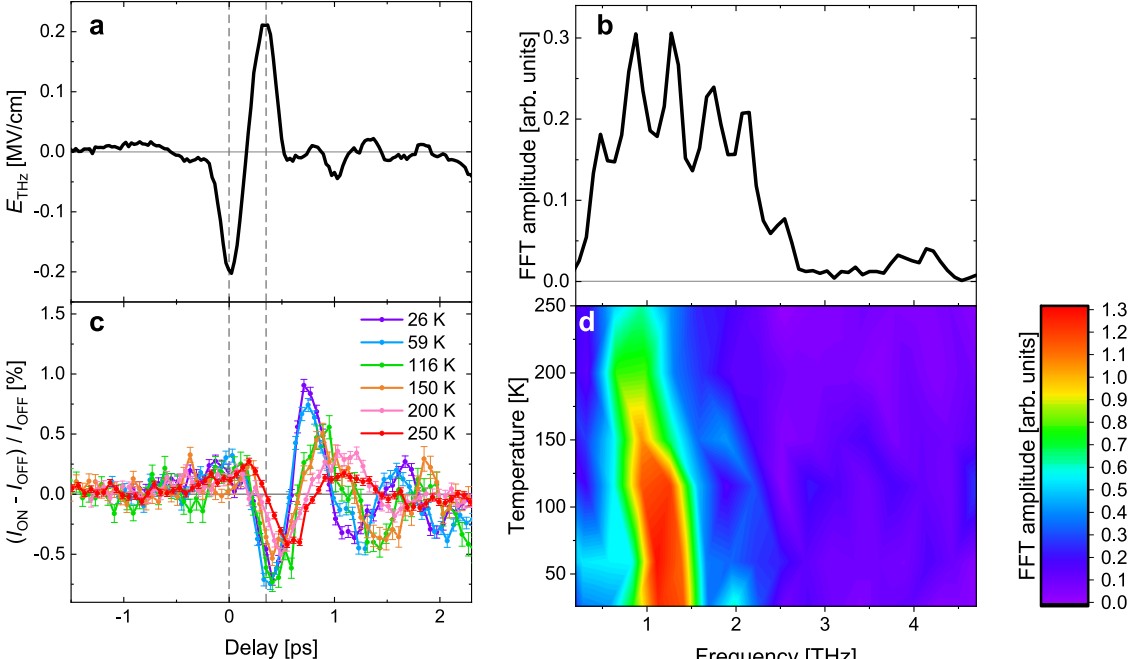

**Fig. 3 | tr-RSXD magnetic scattering signals. a** The incident THz pulse measured with electro-optic sampling at the sample position. **b** THz pulse spectrum. **c** The antiferromagnetic (0 0 4.5) diffraction intensity at various temperatures as a function of pump-probe delay. **d** The FFT spectra of (**c**). Dashed lines in (**a**) and (**c**) are guides for the eyes to visualize the phase shift of π between the incident THz pulse and the magnetic (0 0 4.5) diffraction intensity.

THz pulse (see Methods for detail and Fig. 2b for spectrum with the low-pass filter). The absence of the higher-frequency component in the tr-XRD data taken with the filter (Fig. 2e) and in the FFT of the diffraction intensities (Fig. 2f) fully supports that the higher-frequency mode is resonantly driven by the weak spectral weight around that frequency. As for the low-frequency mode, it has a phase shift of π/2 compared to the THz pulse, consistent with resonantly driven lattice dynamics described by the Lorenz model (see Supplementary Note II). For increasing temperatures, the mode softens with increased damping (Fig. 2f) consistent with the THz-TDS data (Fig. 1d, e). We infer from this that the measured structural dynamics of the mode at -1.7 THz are the structural component of the electromagnon.

One striking feature of our results is the strong difference in the response of the XRD signal of the electromagnon mode when compared to that of the polar vibrational mode at 3.6 THz for data taken without the low-pass filter. As remarked above, even though the spectral power at 3.6 THz in the incident pulse is approximately an order of magnitude weaker than at 1.7 THz, the measured response at 3.6 THz is approximately five times stronger. Without knowing the eigenvector of each mode, it is at present not possible to make a quantitative link between the diffraction response and the atomic displacements associated with each mode. Under the assumption that the (0 0 24) reflection has a similar sensitivity to both modes, however, our results would suggest that a substantial amount of the absorbed THz energy at 1.7 THz is within the spin system. Note that the optical phonon mode is not clearly visible in our THz-TDS data because there is almost no spectral weight around the frequency for a THz pulse used for the measurements (see Supplementary Note III). However, the resonator strength of the electromagnon mode is distinctly larger than that of the optical phonon mode, as can be seen from the previously reported THz reflectivity data taken at low temperatures[18].

### Magnetic sublattice dynamics

Figure 3a,b shows a time trace of the electric field of the incident THz pulse in the tr-RSXD experiment (3a) and its FFT spectrum (3b), respectively, which are similar but with a slightly weaker driving field

strength compared to the tr-XRD experiment. The pump beam polarization was rotated to maximize the field component along [001], as for the tr-XRD experiments. The corresponding time traces of the antiferromagnetic (0 0 4.5) reflection intensity taken at the Fe $L_3$-edge and at various temperatures are shown in Fig. 3c and their FFT spectra in Fig. 3d. This reflection probes the spin-spiral component of the magnetic order[20] (see Supplementary Note IV for a discussion of other negligible contributions) and allows us to use an almost normal incidence angle of the THz pulse (see Methods). The data reveal that intensity variations are about twice as large as those in the structural reflection indicating clear resonantly driven spin dynamics in the hexaferrite. The weakening and redshift for increasing temperature are in qualitative agreement with the behavior of the electromagnon. The obtained low-temperature frequency, however, seems slightly lower in the magnetic response than in the lattice response. Note that previously reported polarized inelastic neutron scattering for a Y-type hexaferrite[17] found magnetic excitation energies consistent with those of their/our THz-TDS data and tr-XRD data, confirming phononic and magnetic frequencies of the electromagnons being likely equal. This is consistent with the fact that the electromagnon mode in the hexaferrite is at the magnetic zone center, in contrast to the electromagnon mode in $R$MnO$_3$[11]. However, due to the small applied magnetic field, the mode could split and could be composed of multiple modes. In addition, the spectral weight of the THz pulse is larger at the low energy range in the tr-RSXD experiments than in the tr-XRD experiments and the probe depth is significantly shorter (see Methods), both of which could slightly modify the response frequency. Another possibility is a systematic error in the measured temperature for this experiment. However, in strong contrast with the tr-XRD data, which uniquely sample lattice dynamics, the oscillations in the magnetic tr-RSXD data exhibit a π phase shift with respect to the THz pulse.

### Discussion

A quantitative assessment of the spin-lattice interaction of the electromagnon requires knowledge of the vibrational and magnetic components in the excitation. For the relative atomic dynamics of the two

modes observed in tr-XRD, we need knowledge of the eigenvector for the modes and/or measurements of additional diffraction peaks. Because of the complicated crystal structure, modeling the vibrational dynamics of the electromagnons in the Y-type hexaferrite on an atomic scale is rather challenging and beyond the scope of this study. However, the oxygen atoms positions located on the borders between the adjacent magnetic blocks are known to dominate magnetic frustration[16], which, in turn, stabilizes the transverse-conical phase shown in Fig. 1b. We expect that even small displacements of these oxygen atoms can significantly affect the magnetic structure. This is consistent with a weak structural electromagnon response in respect to the more general polar vibrational mode. By assuming that the oxygen motions along [001] (with respect to all the rest atoms) represent the vibrational part of the electromagnons in the Y-type hexaferrite, a ~0.32 pm displacement results in the 0.5% change in the (0 0 24) diffraction intensity observed at 20 K (see Supplementary Note V). This amount of displacement changes the bond angles dominating magnetic frustration by ±0.15°, which is large enough to significantly affect the exchange interactions among magnetic moments[21].

The proposed model of the spin dynamics of electromagnons for Y-type hexaferrites, shown in Fig. 1c, gives rise to oscillations of the (0 0 4.5) magnetic diffraction intensity via changes in the two spin canting angles of the L and S blocks as shown in Fig. S6. A quantitative estimation (see Supplementary Note VI) indicates spin excitation amplitudes in the order of those reported for TbMnO$_3$[13], where a rotation of the spin-cycloid plane of ~4° was observed.

From the obtained spin excitation amplitudes, we can estimate the energy required for the maximal moment deflection with an approximation of the effective magnetic Hamiltonian. The energy associated with the lattice excitation could in principle be obtained by a density-functional theory calculation. This, however, requires inclusion of the magnetoelectric coupling, which is extremely challenging. Instead, we can compare the magnetic energy with the absorbed THz energy per formula unit (see Supplementary Note VII + VIII). Our simple estimations show that the magnetic system may acquire a significant fraction of the absorbed THz energy. This suggests that performing measurements like this study is a viable method for obtaining coupling coefficients of the spin-lattice coupling of the electromagnon resonance.

Another interesting observation is the difference in phase shift of the lattice vibration and the magnetic response relative both to each other and to the THz excitation. With respect to the THz pulse, the magnetic mode shows a π phase shift similar to that observed in TbMnO$_3$[13], whereas the phononic part shows a π/2 phase shift, consistent with a resonantly driven damped harmonic oscillator. In the TbMnO$_3$ study, the phase shift was interpreted as a consequence of measuring components of the spin dynamics that comprise the conjugate momentum of an electric-dipole active normal mode consisting of both vibrational and spin components. In a simplified picture, the $E_{THz}$ applies a direct force to the atoms, which causes changes in an effective magnetic field that then drives the spins. This is consistent with the "two-step" model of Ref. 11, where the $E_{THz}$ alters the exchange coupling constants $J$, and the derivative of the Hamiltonian with respect to the spin then creates an effective field that moves the spins. Our observation of the phase shift of the atomic motion relative to the spins thus indicates that electromagnons in the Y-type hexaferrite are type II. Otherwise, type I electromagnons are expected to show in-phase oscillations between the phononic and magnonic parts because type I electromagnons originate from the THz-induced oscillations of static electric polarization produced by magnetic order. Note that, however, this needs to be confirmed by future experiments.

In summary, we have directly observed the coupled dynamics of a coherently driven electromagnon, meaning both the atomic and the spin motions. Although previous work has shown that large-scale coherent magnetic dynamics can be driven by resonant excitation of an electromagnon[13], direct measurements of both the magnetic and the atomic displacement components are needed to fully understand the underlying coupling mechanisms. These are essential for obtaining material control through electromagnon excitations. Our results show that it is possible to establish this important link by using ultrafast X-ray diffraction techniques. Extending it to X-ray magnetic circular dichroism and X-ray magnetic linear dichroism, ultrafast X-ray techniques can be applied to study general materials having spin-lattice coupling even without accessible magnetic diffraction peaks, i.e., while non-resonant X-ray diffraction probes lattice dynamics, resonant X-ray techniques probe spin dynamics. A better understanding of the magnon-phonon coupling is beneficial for optimizing materials for ultrafast control of magnetism. More generally, the ability to selectively probe the components of hybrid excitations is useful in a variety of strong-correlated electron systems where it is important to disentangle strongly coupled degrees of freedom. This can potentially contribute to a better understanding of high-$T_C$ superconductivity[22], topological properties[23–25], spintronics[26,27], and quantum computing[28–31].

## Methods

### Time-resolved X-ray diffraction experiments
The tr-XRD experiments were performed at the Bernina endstation[32,33] of the ARAMIS branch of SwissFEL at the Paul Scherrer Institute (Switzerland). The tr-RSXD experiments were performed at the Furka endstation, which is a new endstation for time-resolved studies in condensed matters at the Athos branch of SwissFEL at Paul Scherrer Institute (https://www.psi.ch/en/swissfel/furka).

Few-cycle phase-stable THz pulses with a central frequency of ~1.7 THz were generated by optical rectification of near-infrared femtosecond laser pulses in an OH-1 crystal. The time-dependent electric field $E_{THz}$ incident on the sample was measured by electro-optic sampling[34] using a [110] oriented GaP crystal with 100 μm thickness and 800 nm pulses copropagating with THz pulses. From the THz-induced polarization rotation angle or phase retardation of 800 nm pulses, one can obtain the absolute scale of $E_{THz}$. Figures 2a and 3a show $E_{THz}$ with the corresponding FFT spectra in Figs. 2b, 3b. The peak $E_{THz}$ strength is ~0.36 MV/cm and ~0.2 MV/cm for the experiments at Bernina and Furka, respectively. For the hard X-ray data shown in Fig. 2e, f, the small high-frequency component above ~3 THz was suppressed by the insertion of a low-pass filter before the sample (compare orange and blue curves in Fig. 2b, spectra with and without the low-pass filter, respectively). The low-pass filter is a multi-mesh filter whose cutoff frequency is 3 THz and transmission below the frequency is more than 85% (QMC INSTRUMENTS Ltd.).

The X-ray photon energy for the hard X-ray experiment was ~7.5 keV, which is slightly above the Fe K edge, with a bandwidth of ~11 eV. For the soft X-ray data, the X-ray photon energy was set to 711 eV, the maxima of an Fe $L_3$ edge X-ray absorption spectrum (see Supplementary Note IX). The energy resolution is ~1 eV. A Y-type hexaferrite single-crystal sample, grown by a flux method described in ref. 35, with a surface parallel to (1 0 16) was used, resulting in an angle of ~30° with the (001) plane. The electric field polarization of the incident THz pulse was adjusted to maximize its component along [001] by rotating the polarization of the near-infrared femtosecond laser and the OH-1 crystal accordingly. A relatively large incident angle of the X-ray beam to the surface, ~58° for the hard X-ray experiment and ~87° for the soft X-ray experiment, results in a grazing exit angle of ~0.5° for the (0 0 24) reflection in the hard X-ray experiment while ~36° for the (0 0 4.5) reflection in the soft X-ray experiment. This leads to an excited probe depth of ~52 nm and ~13 nm for the two experiments (penetration depth of the THz pulse is estimated from Fig. 1d as ~10 μm

at 20 K by assuming a single peak at the resonant frequency in the imaginary part of the refractive index). The sample is mounted together with a pair of permanent magnets (-0.1 T//[010]) to stabilize the multiferroic transverse-conical phase[36] and is cooled with a He-flow cryostat for all X-ray experiments. We used an in-vacuum Jungfrau pixel detector for the hard X-ray experiment to collect both a diffraction peak and fluorescence signals simultaneously and used the latter for the normalization of X-ray intensity fluctuation of the incoming beam. On the other hand, for the soft X-ray experiment, we used an avalanche photodiode to collect diffraction intensities. The normalization was done by measuring the X-ray intensities of diffracted beams by transmission gratings placed upstream, using a photodiode. In all experiments, errors are defined by the standard deviation from multiple scans. The two experimental geometries are displayed in Supplementary Note X.

## Data availability

Experimental data are accessible from the PSI Public Data Repository (https://doi.org/10.16907/3dcbb3d7-aeec-48b7-93f9-3b479f996d5b).

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

## Acknowledgements

Time-resolved hard X-ray diffraction eixperiments were performed at the Bernina endstation of ARAMIS branch of SwissFEL at Paul Scherrer Institute under proposal No. 20210167, and time-resolved soft X-ray diffraction experiments were performed at the Furka endstation of the

ATHOS branch of SwissFEL at Paul Scherrer Institute while in-house beamtime. Before the experiments, the sample was characterized at the Material Science beamline and the X11MA beamline in the Swiss Light Source at Paul Scherrer Institute. We acknowledge B. Pedrini for his support of static characterization. H.U. and V.U. acknowledge the National Centers of Competence in Research in Molecular Ultrafast Science and Technology (NCCR MUST-No. 51NF40-183615) from the Swiss National Science Foundation and from the European Union's Horizon 2020 research H.U. also acknowledges the innovation program under the Marie Skłodowska-Curie Grant Agreement No. 801459—FP-RESOMUS. L.L. and G.S.P. are supported by funding from the Swiss National Science Foundation through Project No. 20021-196964 and No. 200021_169698. E.S. is supported by the NCCR Materials' Revolution: Computational Design and Discovery of Novel Materials (NCCR MARVEL No. 182892) from the Swiss National Foundation and the European Union's Horizon 2020 research and innovation program under the Marie Skłodowska-Curie Grant Agreement No. 884104 (PSI-FELLOW-III-3i). A.N. also acknowledges the Marie Skłodowska-Curie Grant Agreement No. 884104 (PSI-FELLOW-III-3i). E.A. and J.D. acknowledge support from the Swiss National Foundation through Ambizione Grant PZOOP2_179691. This study was in part supported by the Swiss National Science Foundation (SNSF) Requip No. 206021_189640.

## Author contributions

H.U. and U.S. conceived and designed the project. H.U. and T.K. grew single crystals of the Y-type hexaferrite. H.U., J.D., E.A., M.Sav., and S.L.J. performed THz time-domain spectroscopy experiments. H.U., R.M., M.San., Y.D., L.L., A.N., E.S., V.U., G.S.P., J.D., S.G., E.A., H.T.L., S.L.J., and U.S. performed time-resolved hard X-ray diffraction experiment, while H.U., E.P., B.L., C.W., L.P., E.R., and U.S. performed time-resolved soft X-ray diffraction experiment. C.S. contributed to developing the experimental setup at the Furka endstation. H.U. analyzed the experimental data with inputs from R.M. and E.R. Finally, H.U., S.L.J., and U.S. interpreted the results and wrote the manuscript with contributions from all authors.

## Competing interests

The authors declare no competing interests.
