## [Peer Review File · Nature Communications]

REVIEWER COMMENTS

Reviewer #1 (Remarks to the Author):

The authors use time-resolved X-ray diffraction to directly access the spin-lattice coupling by measuring both ultrafast atomic motion and the associated spin dynamics by an intense THz pulse in a multiferroic hexaferrite. This is a unique way to directly observe the combined lattice and magnetization dynamics, which can access the mode-selective spin-lattice coupling strength. Overall, it is a nice work, some important clarifications need to be made before it is possible to recommend it for publication in Nature Communications.

1. The authors mainly focus on the analysis of (0 0 24) reflection intensity in Fig.2 and (0 0 4.5) antiferromagnetic reflection in Fig.3, can authors explain why they specifically study these two peaks? More details can be discussed.

2. In THz time-domain spectroscopy, a clear divergent feature centered at ~ 1.7 THz is identified as an electromagnon, and it has been observed in FFT of the time traces from (0 0 24) reflection intensity. Nevertheless, the observed FFT of the time traces from (0 0 4.5) diffraction intensity is about ~ 1.25 THz, whether this mode is pure magnon? How do you relate this mode to spin-lattice coupling?

3. "Upon heating, the mode shows a red shift concomitant with a weakening of the resonator strength", where the real part n in Fig.1d seems to have disappeared above 200 K, yet the imaginary part k is gradually decreased above 200 K in Fig.1e ?

4. Whether this time-resolved X-ray diffraction is a universal method? Can it be applied to other materials systems with spin-lattice coupling? More discussion can be added.

5. The authors are suggested to comment on the limitations of the approach and how to overcome these possible limitations?

Reviewer #2 (Remarks to the Author):

This manuscript is very interesting because it studies for the first time the effect of resonant THz pumping of an electromagnon on time-resolved X-ray diffraction (tr-XRD), i.e. on crystal and magnetic structure. The authors chose a multiferroic Y-hexaferrite crystal and showed that THz excitation of the electromagnon not only affects the magnetic structure seen in time-resolved resonant soft X-ray magnetic diffraction (tr-RSXD) for antiferromagnetic (0 0 4.5) spot. The similar effect has been demonstrated previously in TbMnO₃ by a similar author team (ref. 13). However, they additionally discovered for the first time that pumping of the electromagnon also causes the temporal oscillations of the crystal diffraction spot (0 0 24) detected by tr-XRD. The magnetic response seen in tr-RSXD has a π phase shift and the structural response has a $\pi/2$ phase shift to the driving THz field. This is a very

challenging and unique experiment and the results are indeed worthy of publication in Nature Commun. However, I have a few questions before final acceptance of the manuscript.

1. If a THz source without filter is used, two peaks are observed in Fig. 2d. The low-frequency peak at 1.5 THz corresponds to an electromagnon and the stronger one at 3.8 THz the authors explain by a polar phonon, which was seen in infrared spectra (ref. 18). However, this phonon is not evident in the THz spectra in Figs. 1d,e. How is this possible? Is it just an inappropriate color scale in Fig. 1e? The Figs. 1d,e. are quite unclear to me, can the authors rather plot the spectra every 50 K to see their temperature evolution? Maybe such spectra could be put into a supplement.

2. The classification of electromagnons into type I and II is unclear to me, it is not known in the literature at all. Do I understand correctly that strong type II electromagnons are activated by magnetostriction, while weak type I electromagnons are activated by DM interaction? If that were the case, then type I electromagnons should be more likely typical for excitation in BiFeO₃ - see e.g. New J. Phys. 18 (2016) 043025 – and not for RMnO₃ (ref. 8). Or do you mean Type I electromagnons those excitations that are activated by the same magnetoelectric coupling mechanism as static ferroelectric polarization? If type II electromagnons cause a phase shift of π in tr-RSXD, do you expect the same or different phase shift for type I electromagnons? I think probably the second statement is correct, but it is not very clear from your text. Please explain it, this is one of the major conclusions of this article.

3. The electromagnon always gets its dielectric strength from the polar phonon to which it is coupled. Is it a phonon at 3.6 THz in Y-hexaferrite?

4. Vit et al. (J. Phys. Soc. Japan 91, 104703 (2022)) tried to observe the nonlinearity of electromagnon absorption under intense THz pumping, which could occur when the magnetic structure in Y-hexaferrite changes. They observed a temperature effect due to heating of the sample by strong THz radiation rather than a change in magnetic structure. Does your experiment show the same effect, i.e. the TC structure does not change during THz pumping, only oscillations of the magnetic structure occur due to dynamical changes of angles for spins in L and S blocks?

6. Minor errors (typo):

Line 16: between different degrees...

Line 59: ...is called electromagnon

Line 118: imaginary part of N is only in Fig. 1e

Line 135: ref. 16 is not about the IR spectra, you probably mean ref. 18

Line 150: consistently x consistent

Line 214: there should be 4 degrees instead of percent

Line 283: you write 92 degrees for the soft X-ray, but in supplement is 2.6 degrees (text just before Eq. 27). Please unify it.

Ref. 8 was published in 2012.

Supplement:

Fig. S2: Please mention that the measurement was performed with THz filter.

Last paragraph on page 10, there should be probably "...up to one-third of Delta with respect to Delta' "

Fig. S8: There is typo on the horizontal axis, there should be "Delta or Delta' "

I like the article very much in general. It may be published after the above points have been explained and minor errors revised.

Reply to the referee report

Let us first thank the referees for the careful reading and the constructive and positive evaluation of our paper.

Below, we give a point-to-point answer to the raised points. Referee comments are given in blue, our reply in black, and changes made to the manuscript are in red.

Referee 1:

We very much appreciate the positive evaluation by this referee, who admits that this is nice work. He/she would like to have some clarifications before giving a recommendation for our manuscript to be published in Nature Communications.

- 1) “The authors mainly focus on the analysis of (0 0 24) reflection intensity in Fig.2 and (0 0 4.5) antiferromagnetic reflection in Fig.3, can authors explain why there specifically study these two peaks? More details can be discussed.”

The (0 0 4.5) magnetic reflection was chosen because (1) it directly originates from the multiferroic order (spin-spiral component) (see Ref. 20) and (2) it allows us to use a close to normal incidence angle on the surface of the nearly-collinear THz and x-ray beams. An incidence angle of THz close to the normal incidence “minimizes” the footprint and maximizes the effective THz-field strength in the probed region. Therefore, we can strongly drive the mode.

The (0 0 24) Bragg reflection was chosen because of multiple reasons. (1) It allowed us to employ a large incidence angle (THz + X rays), similar to the (0 0 4.5) magnetic reflection, and a grazing exit angle (X rays). Although the penetration depth of the THz beam is fairly large as mentioned in Methods, it is still smaller than for the X rays. Grazing exit limits the probing depth of the X-ray beam to the fully pumped region. (2) The (0 0 24) Bragg reflection intensity is only sensitive to atomic motions along [001]. The electromagnon mode is IR-active for THz electric field along [001], and we expect this to be the main direction of atomic motions. (3) The (0 0 24) reflection also has a relatively small structure factor due to the destructive interference of X rays scattered from different atoms, which enhances its sensitivity (relative change of intensity with respect to a given motion) to tiny atomic motions.

We added the following sentences in the main text to clarify the reasons why we chose these reflections.

“The (0 0 24) reflection was chosen because it is exclusively sensitive to atomic motions along [001] and it allows us to use a large incidence angle of a collinear THz and X-ray pulses to the sample surface, resulting in a large E_{THz} inside the sample. This reflection also makes possible a grazing exit angle of 0.5° that results in a short probing depth which is more homogeneously pumped (See Methods for the penetration depths of the beams). Furthermore, with respect to a given atomic motion, the relative change of diffraction intensities is generally large for a reflection with a small structure factor, as is the case for the (0 0 24) reflection. Different reflections will therefore also be sensitive to additional modes. (Supplementary Note I)” on p. 4-5.

“This reflection probes the spin-spiral component of the magnetic order [20] (see Supplementary Information Note. IV for a discussion of other negligible contributions) and allows us to use an almost normal incidence angle of the THz pulse (see Methods).” on p. 7.

- 2) “In THz time-domain spectroscopy, a clear divergent feature centered at ~ 1.7 THz is identified as an electromagnon, and it has been observed in FFT of the time traces from (0 0 24) reflection intensity. Nevertheless, the observed FFT of the time traces from (0 0 4.5) diffraction intensity is about ~ 1.25 THz, whether this mode is pure magnon? How do relate this mode to spin-lattice coupling?”

The obtained frequency from the oscillations of the (0 0 4.5) magnetic reflection intensity is indeed lower than those obtained from the THz time-domain spectroscopy and the oscillations of the (0 0 24) reflection intensity, as the referee pointed out. We believe that the oscillations of the magnetic reflection intensity are not due to the excitation of a pure magnon but rather the electromagnon mode because of the following reasons, which we discuss on p. 7 in the main text.

First, Mochizuki et al. pointed out that the resonant frequency observed in THz time-domain spectroscopy (electric-dipole active part, i.e., lattice) and the resonant frequency observed in polarized inelastic neutron scattering (magnetic part) can be different for a particular branch of electromagnons in $RMnO_3$ because of the dispersion of the mode [11]. Polarized inelastic neutron scattering probes the mode at the magnetic zone center q while the THz time-domain spectroscopy probes the mode at a particular point where the mode is electric-dipole active, which is $\pi-2q$. Hence, the dispersion gives different resonant frequencies for these techniques. In our case, however, the mode is electric-dipole active at the magnetic zone center, described by the magnetic propagation vector (0, 0, 1.5) (see Fig. 1c). Therefore, one expects that the techniques employed in this study should not give different resonant frequencies. Actually, Ref. 17 reported a polarized inelastic neutron scattering study combined with THz time-domain spectroscopy on a very similar Y-type hexaferrite as ours. They observed a clear infrared-active mode that is ascribed to electromagnons by THz time-domain spectroscopy, and its resonance frequency is almost the same as the electromagnon mode in our sample. Their polarized inelastic neutron scattering data shows magnetic excitations at the same energy as the mode determined by the THz time-domain spectroscopy. Therefore, the magnetic excitations are infrared-active, as they should be for electromagnons per definition. In other words, the resonance frequency of the phononic part likely equals the resonance frequency of the magnetic part. Besides, the temperature dependence shown in Fig. 3d shows a similar trend as the THz time-domain spectroscopy data (Fig. 1e) and the hard X-ray data (Fig. 2f). Furthermore, if the oscillations in the magnetic diffraction intensity are due to pure magnons that are somehow resonantly excited by a THz pulse possibly via magnetic-dipole excitations, this mode should still be observed by THz time-domain spectroscopy, which is not the case. Based on these considerations and observations, we believe the oscillations in the magnetic diffraction intensity are ascribed to electromagnons, which arise via spin-lattice coupling.

Then a question is why the obtained frequency from the magnetic diffraction intensity oscillations is different from the other two measurements. Possible reasons already discussed in the main text are;

- (1) Slightly different applied magnetic field because it is known from Ref. 17 that the electromagnon resonance frequency depends on applied magnetic field strength.
- (2) The more surface sensitivity of the magnetic diffraction measurements than the other two measurements because of the shorter probing depth, which may cause the soft X-ray data to be affected by surface strain possibly created during sample preparation.
- (3) More THz spectral weight in the lower frequency component for the THz pulse used for the soft X-ray experiment, compared to the THz pulse used for the hard X-ray experiment.

In addition to them, it may be possible that the temperature for the soft X-ray experiments has a systematic error and is actually higher than those shown in the manuscript.

We added the following sentences on p. 7.

“This is consistent with the fact that the electromagnon mode in the hexaferrite is at the magnetic zone center, in contrast to the electromagnon mode in $RMnO_3$ [11].”
 “Another possibility is a systematic error in the measured temperature for this experiment.”

- 3) ““Upon heating, the mode shows a red shift concomitant with a weakening of the resonator strength”, where the real part n in Fig.1d seems directly disappeared above 200 K, yet the imaginary part k is gradually decreased above 200 K in Fig.1e ?”

Below shows extracted n and k at 200 K and 250 K from Figs. 1d and 1e, respectively. The mode remains both in n and k .

As seen in reply to the first comment from the second referee, the THz spectrum for the THz time-domain spectroscopy experiment does not contain frequency components above 3 THz unlike one for time-resolved X-ray experiments. Thus, we cut the data above 2.5 THz from Figs. 1d and 1e. In addition, we corrected the color

scale in Figs. 1d and 1e because the color scale showed white for high values of n or k , which is not the case as written in the caption.

- 4) “Whether this time-resolved X-ray diffraction is a universal method? Can it be applied to other materials systems with spin-lattice coupling? More discussion can be added.”
- 5) “The authors are suggested to comment on the limitations of the approach and how to overcome these possible limitations?”

We appreciate these questions/suggestions. We combine the answer to these two questions as they are related.

One of the limitations of this technique is the accessibility of X-ray magnetic scattering, which in the soft X-ray regime is limited because of given edges requiring relatively small photon energy (small momentum transfers). This can be overcome by using hard X rays, though the scattering signals are very weak at transition-metal K edges and might limit the observation of a signal in some cases. The other limitation is the pump spectrum, which requires a large frequency component at the mode resonance. However, a significant development in the laser field has enabled us to access a quite wide frequency range of intense THz/mid-IR pulses. It also requires the materials to be insulating, so that the THz radiation is absorbed into a mode but not into the electronic system. One could also use XMCD/XMLD as a probe for the magnetism. Although there are some limitations as mentioned above, therefore, this time-resolved X-ray diffraction technique is a universal approach.

We think it would go beyond this article to get too much into the technicality of how these approaches can be further improved.

We modified and added the following sentences in the last paragraph of the main text.

“Our results show that it is possible to establish this important link by using ultrafast X-ray diffraction techniques. **Extending it to X-ray magnetic circular dichroism and X-ray magnetic linear dichroism, ultrafast X-ray techniques can be applied to study general materials having spin-lattice coupling even without accessible magnetic diffraction peaks, i.e., while non-resonant X-ray diffraction probes lattice dynamics, resonant X-ray techniques probe spin dynamics.**”

Referee 2:

Again, we very much appreciate the positive evaluation by this referee, who judges our paper as worthy of publication in Nature Communications. We are very glad to read the comment, “I like the article very much in general. It may be published after the above points have been explained and minor errors revised.”.

- 1) “If a THz source without filter is used, two peaks are observed in Fig. 2d. The low-frequency peak at 1.5 THz corresponds to an electromagnon and the stronger one at 3.8 THz the authors explain by a polar phonon, which was seen in infrared spectra (ref. 18). However, this phonon is not evident in the THz spectra in Figs. 1d,e. How is this possible? Is it just an inappropriate color scale in Fig. 1e? The Figs. 1d,e. are quite unclear to me, can the authors rather plot the spectra every 50 K to see their temperature evolution? Maybe such spectra could be put into a supplement.”

The absence of the optical phonon mode at ~ 3.6 THz in the THz time-domain spectroscopy data is due to a different spectrum of the THz pulse used for the measurements, which has almost no spectral weight around this frequency, compared to the one used for the time-resolved hard X-ray measurements. The difference in their spectra could be due to (1) different pulse duration of near-infrared lasers for optical rectification (100 fs for the time-resolved X-ray experiments, while 100 fs – 120 fs for the THz time-domain spectroscopy experiment; both in the full width at half maximum) and/or (2) different thickness of OH-1 crystals because of strong absorption around the high-frequency components (500 μm for time-resolved X-ray experiments while 700 μm for the THz time-domain spectroscopy experiment).

As we mentioned in reply to the third question given by the first referee, we cut the data above 2.5 THz. We added a new section in Supplementary Information to show a spectrum of the THz pulse for the THz time-domain spectroscopy and the following sentence on p. 6 in the main text.

“Note that the optical phonon mode is not clearly visible in our THz-TDS data because there is almost no spectral weight around the frequency for a THz pulse used for the measurements (see Supplementary Information Note III). However, the resonator strength of the electromagnon mode is distinctly larger than that of the optical phonon mode, as can be seen from the previously reported THz reflectivity data taken at low temperatures [18].”

IX. The spectrum of THz pulse for THz-TDS measurements

Figure S9 shows the THz pulse spectrum used for the THz-TDS experiment. In contrast to those used for the ultrafast X-ray diffraction experiments, there are almost no components above 3 THz.

Fig. S9 | THz pulse spectrum used for the THz-TDS experiment.

- 2) “The classification of electromagnons into type I and II is unclear to me, it is not known in the literature at all. Do I understand correctly that strong type II electromagnons are activated by magnetostriction, while weak type I electromagnons are activated by DM interaction? If that were the case, then type I electromagnons should be more likely typical for excitation in BiFeO₃ - see e.g. New J. Phys. 18 (2016) 043025 – and not for RMnO₃ (ref. 8). Or do you mean Type I electromagnons those excitations that are activated by the same magnetoelectric coupling mechanism as static ferroelectric polarization? If type II electromagnons cause a phase shift of π in tr-RSXD, do you expect the same or different phase shift for type I electromagnons? I think probably the second statement is correct, but it is not very clear from your text. Please explain it, this is one of the major conclusions of this article.”

The referee correctly understands the classification of electromagnons into two types. We label them here as type I and type II. Type I electromagnon we define being due to the oscillation of static electric polarization and is called as “cross-coupling electromagnon” in Ref. 8. On the other hand, type II electromagnon we define being due to magnetostriction and is called as “magnetostriction-induced electromagnon” in Ref. 8. This classification is introduced in Ref. 8, and $RMnO_3$ (R = rare-earth) shows these types of electromagnons.

While type II electromagnons are due to the creation of new electric-dipole moments via the two-step model (Ref. 11), type I electromagnons are due to the oscillations of static electric polarization. Hence, we expect in-phase oscillations of phononic and magnonic parts for type I electromagnons. We added the following sentences in the main text to clarify the different electromagnon types and to show what observation is expected for type I electromagnons.

“We call this type of electromagnons type I, following the nomenclature introduced by Takahashi et al. [8].”,

“In these “type II” electromagnons [8] the polar lattice distortion modifies exchange interactions...”,

and

“Electromagnons were originally envisioned as excitations arising from the coupling between magnetic order and the crystal lattice that gives rise to the equilibrium electric polarization, i.e., THz-induced oscillations of static electric polarization produced by magnetic order [7].” on p. 2.

“Otherwise, we expect that type I electromagnons show in-phase oscillations between the phononic and magnonic parts because type I electromagnons originate from the THz-induced oscillations of static electric polarization produced by magnetic order. Note that, however, this needs to be confirmed by future experiments.” on p. 9.

- 3) “The electromagnon always gets its dielectric strength from the polar phonon to which it is coupled. Is it a phonon at 3.6 THz in Y-hexaferrite?”

Since we did not measure the temperature dependence of the (0 0 24) reflection without the low-pass filter and the spectral weight of a THz pulse for THz time-domain spectroscopy around 3.6 THz is negligible (see our reply to the first point raised by this referee), we unfortunately do not know if the optical phonon mode at 3.6 THz is the mode coupling to the electromagnon mode. However, as an electromagnon mode is typically coupled to the lowest-energy phonon mode in RMnO_3 [for example, *Phys. Rev. B* **74**, 100304(R) (2006)], it is possible that this optical phonon mode couples to the electromagnon mode.

- 4) “Vit et al. (*J. Phys. Soc. Japan* 91, 104703 (2022)) tried to observe the nonlinearity of electromagnon absorption under intense THz pumping, which could occur when the magnetic structure in Y-hexaferrite changes. They observed a temperature effect due to heating of the sample by strong THz radiation rather than a change in magnetic structure. Does your experiment show the same effect, i.e. the TC structure does not change during THz pumping, only oscillations of the magnetic structure occur due to dynamical changes of angles for spins in L and S blocks?”

We do not believe (average) heating could be critical in our measurements because the repetition rate is significantly different between our experiment (100 Hz) and the experiment shown done by Vit et al. (200 kHz). Considering heating within a single pulse, if prominent, we should have observed the redshift of the electromagnon in the time trace. Namely, the periodicity of the oscillations in time-resolved X-ray diffraction should become longer at a later delay time. However, such a change is not observed.

We also corrected all the typos that the referee pointed out.

REVIEWERS' COMMENTS

Reviewer #1 (Remarks to the Author):

The authors have addressed all of the comments presented by the reviewers. They also modified the paper accordingly. I would recommend to publish the manuscript and I believe the authors will carefully proofread the information in the text.

Reviewer #2 (Remarks to the Author):

The authors have edited the manuscript according to the comments of both reviewers. I have no more comments on the text. The paper is very well written and is now suitable for publication in Nature Communications.